# Stability of *N*-Acetylcysteine (NAC) in Standardized Pediatric Parenteral Nutrition and Evaluation of *N*,*N*-Diacetylcystine (DAC) Formation

**DOI:** 10.3390/nu12061849

**Published:** 2020-06-21

**Authors:** Isabelle Sommer, Hervé Schwebel, Vincent Adamo, Pascal Bonnabry, Lucie Bouchoud, Farshid Sadeghipour

**Affiliations:** 1Department of Pharmacy, University Hospital of Lausanne, 1011 Lausanne, Switzerland; farshid.sadeghipour@chuv.ch; 2Institute of Pharmaceutical Sciences of Western Switzerland, University of Geneva, University of Lausanne, 1211 Geneva, Switzerland; pascal.bonnabry@hcuge.ch; 3B. Braun Medical AG, 1023 Crissier, Switzerland; herve.schwebel@bbraun.com (H.S.); vincent.adamo@bbraun.com (V.A.); 4Department of Pharmacy, Geneva University Hospitals, 1205 Geneva, Switzerland; lucie.bouchoud@hcuge.ch

**Keywords:** parenteral nutrition, pediatrics, amino acids, cysteine, *N*-acetylcysteine, *N*,*N*-diacetylcystine

## Abstract

The ESPGHAN/ESPEN/ESPR-Guidelines on pediatric parenteral nutrition (PPN) recommend the administration of the semiessential amino acid (AA) cysteine to preterm neonates due to their biochemical immaturity resulting in an inability to sufficiently synthetize endogenous cysteine. The soluble precursor *N*-acetylcysteine (NAC) is easily converted into bioavailable cysteine. Its dimer *N*,*N*-diacetylcystine (DAC) is almost unconvertable to cysteine when given intravenously resulting in a diminished bioavailability of cysteine. This study aims to understand the triggers and oxidation process of NAC to DAC to evaluate possibilities of reducing DAC formation in standardized PPN. Therefore, different air volumes (21% O_2_) were injected into the AA compartment of a standardized dual-chamber PPN. O_2_ concentrations were measured in the AA solution and the headspaces of the primary and secondary packaging. NAC and DAC concentrations were analyzed simultaneously. The analysis showed that O_2_ is principally delivered from the primary headspace. NAC oxidation exclusively delivers DAC, depending on the O_2_ amount in the solution and the headspaces. The reaction of NAC to DAC being containable by limiting the O_2_ concentration, the primary headspace must be minimized during manufacturing, and oxygen absorbers must be added into the secondary packaging for a long-term storage of semipermeable containers.

## 1. Introduction

The new ESPGHAN/ESPEN/ESPR Guidelines on pediatric parenteral nutrition (PPN) published in 2018 state “Standard parenteral nutrition (PN) solutions should generally be used over individualized PN solutions in the majority of pediatric and newborn patients, including very low birth weight (VLBW) premature infants” [1] (p. 2). Beneath other recommendations concerning the constitution of standardized and individual PPN, these guidelines, as well as the previous ones from 2005, recommend the administration of bioavailable cysteine to preterm neonates [2].

Normally, in healthy adults and infants as well as term newborns, cysteine is synthetized from the essential amino acid (AA) methionine, which serves as donor of sulfur (S-donor), and the nonessential AA serine, which serves as donor of carbon fragments (C-donor). The recommendation for supplementation of cysteine is given due to the biochemical and metabolic immaturity of the premature patients resulting in an inability to sufficiently synthetize endogenous cysteine [3]. Therefore, the so-called semiessential or conditionally essential AA cysteine becomes an essential one [2,4].

Cysteine, having a good solubility, can be administrated intravenously as its hydrochloride salt (cysteine-HCl). The problem is its short stability in solution. Consequently, it should be added to the PPN at the end of administration [2]. Nevertheless, some pharmaceutical companies use cysteine-HCl for their pediatric AA solutions, such as Primene^®^ 10% and Numeta^®^ Neo (Baxter International, Deerfield, IL, USA) or TrophAmine^®^ 10% (B. Braun Medical Inc., Bethlehem, PA, USA), containing 1.89 g/L, 1.13 g/L, and 0.16 g/L of cysteine, respectively. The cysteine concentrations contained in the products of Baxter are quite higher than the recommendations of the ESPGHAN/ESPEN/ESPR Guidelines.

Much more stable in solution but sparingly soluble is the oxidized form cystine, which is composed of two molecules of cysteine. Due to its poor solubility, it is unusable for PPN [2].

A soluble precursor is *N*-acetylcysteine (NAC), the acetylated form of cysteine [5]. In vivo, it is easily converted into bioavailable cysteine after catalysis by an acylase [6,7]. Therefore, cysteine is commonly supplemented by means of administration of a PPN for preterm infants containing an amino acids (AA) mixture including 0.7 g/L of NAC (equivalent to 0.52 g/L of cysteine) (Aminoven^®^ Infant 10%, Fresenius Kabi AG, Germany; Aminoplasmal^®^ Paed 10%, B. Braun Medical AG, Germany; Aminopaed^®^ 10%, Baxter International, Deerfield, IL, USA).

Unfortunately, the bioavailability of NAC is only 50% [8]. “The 2006 Cochrane analysis indicated that Plasma levels of cysteine were significantly increased by cysteine supplementation but not by *N*-acetylcysteine supplementation.” [3] (p. 3).

Another disadvantage of the precursor NAC is that it reacts in the presence of oxygen (O_2_) during the sterilization process and upon storage to its dimer *N*,*N*-diacetylcystine (DAC). DAC is soluble and more stable in PPN solution than NAC. However, it is converted 30 times slower into cysteine when given intravenously than after oral administration where DAC is reduced by cystine-reductase during its passage through the small intestine [6].

Safety, pharmacology, and toxicology studies on DAC in animals revealed no findings that would prevent further clinical evaluation of DAC in animals or humans [9,10]. No other publication treated the topic of DAC safety and/or toxicity.

In order to conform to the ESPGHAN/ESPEN/ESPR recommendation for the use of standardized PPN [1], a working group of pharmacists, neonatologists, and an industrial partner was established. This collaboration aims to develop a standardized PPN as a dual-chamber infusion bag for newborn and preterm infants. Today, such PPN are mostly compounded individually from several sterile raw solutions (AA, glucose, and different electrolytes) in single-chamber containers representing a preparation at risk with short stability. Only a few standardized PPN exist for newborn and especially for preterm infants, like Numeta^®^ Neo (Baxter International, Bethlehem, PA, USA) or Pediaven^®^ AP-HP (Fresenius Kabi AG, Bad Homburg, Germany) for the French market only. However, these formulations do not always conform to therapeutic protocols of all hospitals. Therefore, this standardized PPN represents an alternative solution for nutritional care for newborn and premature infants with a high product quality, long stability, and easy storage conditions.

The development process of this new standardized PPN includes, amongst others, stability testing of all components (AA, glucose, and electrolytes) during storage in the final packaging where analysis of the different AA components showed an increasing presence of DAC, the degradation product of NAC. In conclusion, a decreasing concentration of NAC can also be observed, meaning a diminished availability of cysteine for the treatment of the concerned patients. 

Due to the fact that no safety and toxicity issues are reported for DAC, the degradation limit for active ingredients of 10% was fixed for the product specification, meaning a maximum of 10% of NAC may be lost through oxidation to DAC.

For these reasons, the aim of this study was to find and understand triggers of the chemical reaction of NAC to DAC in PPN to reduce and limit its formation.

## 2. Materials and Methods 

For a long-term stability of the final product, a 250 mL dual-chamber infusion bag was used to separate amino acids from glucose and electrolytes to avoid the Maillard reaction of AA in presence of glucose [11]. The material used for the primary packaging was a B. Braun multilayer coextruded film with barrier properties for oxygen. The secondary packaging consisted of the high gas barrier material silicon dioxide SiOx foil made of cast polypropylene outer and inner layers and a SiOx-coated polyethylene terephthalate middle layer.

The AA compartments of 16 dual-chamber infusion bags were filled with 79 mL of oxygen-saturated AA solution (Aminoplasmal^®^ Paed 10%, B. Braun Medical AG, Melsungen, Germany). As analyses are focused on the AA compartment, the glucose and electrolyte compartments were filled with 171 mL of water for injection. The filling volumes of the dual-chamber infusion bag corresponded to the developed final product, meaning 79 mL of AA solution and 171 mL of a mixture containing glucose and electrolytes. By means of a syringe, residual air from the filling process was withdrawn from the 16 compartments of AA. Afterwards, exact volumes of air (21% of O_2_) were reinjected. The amounts of O_2_ added in the primary headspace (HS1) of the AA compartments were 2 mL = 18.7 µmol (*n* = 6 infusion bags), 8 mL = 75.0 µmol (*n* = 8 infusion bags), and 16 mL = 150.0 µmol (*n* = 2 infusion bags). Before the terminal sterilization process (F_0_ ≥ 15 min, 121 °C), all bags were packaged in the secondary packaging without or with two fast-acting oxygen absorbers SS-500BXMBC with air absorption capacity greater than 500 mL from Mitsubishi (*n*_2mL_ = 3, *n*_8mL_ = 4, and *n*_16mL_ = 1 for each packaging configuration). Most of the headspace volume being cumulated at the port systems, the first oxygen absorber was placed next to the AA port system and the second one next to the port system of the glucose and electrolyte compartment as shown in Figure 1.

The NAC and DAC concentrations were measured by high performance liquid chromatography (HPLC) based on the method described in the European Pharmacopeia [7] and validated by B. Braun Medical for their AA solution Aminoplasmal^®^ Paed 10%.

A variance analysis was determined in accordance with ISO 5725 from three series of six analyses by varying operating conditions (day, analyst, and instrument). The analyses were performed using a homogeneous amino acid mixture sample spiked at around 15 mg/L of DAC in the analytical solution.

The intermediate precision variance, corresponding to the sum of repeatability variance and intergroup variance, was calculated with NeoLiCy^®^ software, version 2.1. Expressed as relative standard deviation, the value of intermediate precision was expected to be <5%.

The same is applicable for the standard deviation of the beta expectation tolerance interval when using the same set of values. According to ISO 21748, this value can be associated to the standard measurement uncertainty. With the experimental design, the number of degrees of freedom (ν) was calculated using the Welch–Satterthwaite approximation.

This validated method was carried out with an Atlantis™ dC_18_ column (4.6 mm × 150 mm, 3 µm). A solution of ammonium formate at 5 mM was used as mobile phase, the flow rate was fixed at 0.7 mL/min and the oven temperature was set to 40 °C. Precisely, 5 µL of the prepared sample solution was injected. Detection was performed by a UV-detector at 210 nm wavelength. All 16 packaged infusion bags were stored at room temperature and protected from light. For each configuration—without or with two oxygen absorbers—8 bags were tested (*n*_2mL_ = 3, *n*_8mL_ = 4, and *n*_16mL_ = 1). The NAC and DAC concentrations as well as the O_2_ amount were analyzed once at each of the seven time points on day 0–before and right after injection of air and at 2.5 h, on days 7, 13, 46, and 95.

Fifteen concentration analyses of the abovementioned bags were randomly chosen to perform a mass balance analysis for NAC and DAC aiming to show any other source of DAC formation or any other degradation product of NAC. Minimum one of each bag configuration concerning storage time (7, 13, or 46 days), air volume (2, 8, and 16 mL), and presence of oxygen absorber (with or without) were chosen to represent all possibilities of final product conception. The results were illustrated as molar percentage of the initial NAC amount present in the AA solution.

For the HPLC analyses, five NAC standard solutions of different concentrations (75, 90, 100, 110, and 120 mg/L) and one reference solution of 100 mg/L were prepared from a stock solution of 1000 mg/L. Five DAC standard solutions (8, 20, 30, 45, and 60 mg/L) and one reference solution of 30 mg/L were prepared from a stock solution of 100 mg/L. Each of these standard solutions were measured twice for the concentration determination of the component to be analyzed (NAC or DAC).

The O_2_ concentration was measured by means of an optic fiber probe (type NTH-PSt7, PreSens) to be inserted with a needle or an optic patch sensor (type SP-PSt8-YAU, PreSens) attached to the inner surface of the primary and secondary packaging. Data were captured and converted using the oxygen-meter Microx 4 from PreSens. The O_2_ concentration was measured in the headspace of the secondary packaging, the headspace of the AA compartment, and directly in the AA solution. This analysis was performed once on all 16 infusion bags and at each of the seven time points (on day 0–before and right after injection and at 2.5 h, on days 7, 13, 46, and 95). The focus was put on the development of O_2_ and DAC concentrations within the HS1 and the solution. Data including error bars were presented in graphs for the different configurations.

The O_2_ consumption was calculated as a ratio of the O_2_ and DAC concentrations throughout the storage duration. The correlation of DAC formation from NAC was evaluated by comparing the concentrations along the test duration.

## 3. Results

The validation of the HPLC method was conform to the fixed specifications. Considering the preparation factor of the method (dilution 2:1), the precision of the method is characterized for levels of DAC at about 0.030 g/L in the AA solution. The Welch–Satterthwaite approximation was 2.84, which gives a coverage factor of 3.4 (Student *t* quantile for ν with 95% of confidence). With this coverage factor, the expended uncertainty can be assumed to be 14.6%.

The concentration analysis performed for NAC and DAC showed that DAC is exclusively delivered by oxidation of NAC. At the same time, this finding also implies that NAC only degrades to DAC and no other side products. The corresponding mass balance resulted in 100% ± 1% for all different storage and manipulation configurations as shown in Figure 2.

Additionally, as shown in Figure 3, DAC formation was directly correlated to the O_2_ consumption (*R*^2^ = 0.9972) and the slope value obtained indicated that 1 molecule of O_2_ forms 1.2 molecules of DAC.

The analysis also showed that HS1 represents the most important contributor of O_2_ available for the chemical reaction of NAC to DAC. Logically, the higher the volume of HS1, the higher its O_2_ amount. Contrary to this result, the available O_2_ amount within the solution remains unchanged (22.2 µmol) because of a dynamic equilibrium between HS1 and solution. The total available O_2_ amount depending on different HS1 volumes are illustrated in Figure 4.

In the absence of oxygen absorbers in the secondary packaging, the headspaces of the primary (HS1) and secondary packaging (HS2) are saturated with O_2_ (~21%). Therefore, the oxidation of NAC to DAC takes place without limit and the DAC concentration raises constantly during the study duration. Due to the dynamic equilibrium between solution and HS1, even after 95 days of storage duration, there is still enough O_2_ available for the formation of DAC from NAC, meaning that this reaction will last until exhaustion of DAC or O_2_. Figure 5 shows that the O_2_ concentrations do not decrease significantly during storage (light symbols) even though DAC concentration raises constantly amongst time (solid symbols). No difference in DAC formation quantity or velocity was observed between the different air volumes (2, 8, and 16 mL) within HS1 (three superposed DAC concentration curves). Error bars are too small to be visible. There was no need to perform other statistical comparison.

The results of the same analysis performed in the presence of two oxygen absorbers showed a considerable decrease of O_2_ concentration (30% in 2.5 h) within the primary packaging headspace (HS1) of the AA compartment, which limits DAC formation considerably (Figure 6). Error bars were present for every curve except for the 16 mL HS1 as only one bag was analyzed. The precision of results was acceptable, error bars never overlapped, so no other statistical testing was performed.

The data showed that once all available O_2_ molecules reacted with NAC, no further DAC was formed. The higher the initial O_2_ concentration, the longer it took to reach the DAC plateau.

With one oxygen absorber having an oxygen absorption capacity of 500 mL, the O_2_ concentration reduction after 24 h was 38 ± 3% (95% confidence intervals 31; 45%) and 42 ± 5% (95% confidence intervals 35; 49%) for dual-chamber infusion bags packaged with one and two oxygen absorbers, respectively (Figure 7). No difference was observed between the dual-chamber infusion bags packaged with one or two oxygen absorbers, *p* = 0.28.

## 4. Discussion

The ESPGHAN/ESPEN/ESPR Guidelines on pediatric parenteral nutrition (PPN) published in 2018 recommend the use of standardized PN for pediatric patients including newborn and very low birth weight premature (VLBW) infants instead of individual PPN [1]. This recommendation considers individual compounded PN to be high-risk preparations due to its high amount of different ingredients and the accompanying multiple manipulation steps. The ESPGHAN/ESPEN/ESPR Guidelines also recommend the administration of the semiessential amino acid cysteine [2,4] to newborn and preterm infants, but they do not mention the best source to be used (cysteine-HCl or *N*-acetylcysteine) [2,3]. Due to its good solubility and stability in PPN and its ability to be easily transformed into cysteine [6,8], most amino acid mixtures for pediatric patients contain *N*-acetylcysteine (NAC) although its bioavailability is of only 50% [3].

It is known that NAC is oxidized to its dimer *N*,*N*-diacetylcystine (DAC) [6,7]. When given orally, this reaction is easily reversible as DAC is reduced to NAC and further to cysteine during its passage through the small intestine. This is the missing step when NAC is administered intravenously [6,8]. Therefore, oxidized NAC stays unavailable for cysteine delivery in PPN.

The problem with the degradation product DAC in PPN seems to be quite a new one. No publication was found despite the study of Böhler from 1988 [6] treating the product DAC in parenteral nutrition. As DAC is tested in mice for oral treatment of atherosclerosis, for instance, there is no evidence of toxicity or nonsecurity of DAC in animals [9], but this has never been subject to research in humans.

The development process of a standardized PPN in a dual-chamber infusion bag for the first days of life of a newborn and/or premature infant revealed the problem of oxidation of NAC to DAC, which has never been described as having any toxicological consequences on the treated patients [10]. Nevertheless, profound examinations of this chemical reaction have been conducted. 

The results of this study show that DAC is exclusively generated from NAC in presence of oxygen (O_2_). The mass balance of NAC and DAC concentrations being 100% excluded the formation of other degradation products of NAC. These facts lead to the conclusion that the focus on available O_2_ is the most important point to limit this reaction.

We also demonstrated that the headspace of the primary packaging of the dual-chamber infusion bag is the most important source of O_2_. Therefore, this volume of potential O_2_ donor needs to be reduced to a minimum. A headspace volume of 2–8 mL for the AA compartment volume of 79 mL seems to be the most adapted and feasible way to limit the oxidation of NAC to DAC. Additionally, one oxygen absorber must be placed in the secondary packaging next to the port system of the AA compartment, where most of the air is cumulated. This helps to maximally reduce the O_2_ concentration within the primary headspace and the AA solution. Indeed, our bags have a multilayer coextruded film which is still little permeable for oxygen. A second oxygen absorber may be placed next to the port system of the glucose and electrolyte compartment to help reduce the O_2_ concentration within this compartment as well.

For standardized parenteral nutrition solutions as well as for individual compounded PN, the total amount of O_2_ present in the system should generally be kept at a minimum level to prevent any possible oxidation. Our study’s results suggest two interesting future research projects. First, finding of the definition of an acceptable limit for the transformation of NAC to DAC, knowing that the available concentration of the semiessential amino acid cysteine reduces at the same time. It is not known whether the recommended concentration of cysteine for PPN published by the ESPGHAN/ESPEN/ESPR Guidelines [2] takes this reaction into account. Second, carrying out a retrospective pharmacovigilance study to evaluate the impact of this chemical reaction, its resulting diminished bioavailability of cysteine, as well as the presence of DAC in the newborn and preterm infants.

## 5. Conclusions

There is no official recommendation of which source to use for the supplementation of cysteine in standardized or individual pediatric parenteral nutrition for newborns including very low birth weight preterm infants. The main precursor of cysteine used is *N*-acetylcysteine. It is soluble in PPN solutions, but easily oxidized to its dimer *N*,*N*-diacetylcystine in the presence of oxygen. To limit this reaction and the resulting diminished bioavailability of cysteine in PPN solutions, the concentration of oxygen needs to be reduced to a minimum in the primary headspace of the amino acid solution compartment of a dual-chamber infusion bag. This can be achieved by limiting the headspace volume in the amino acid compartment and by adding performant oxygen absorbers into the secondary packaging.

For our developed standardized PPN, based on the results obtained, we decided to reduce the air volume of the primary headspaces to a minimum (<8 mL) during the semi-automated filling process. To reduce the residual oxygen amount in the system and the resulting *N*,*N*-diacetylcystine formation, the filled dual-chamber infusion bag is packaged as soon as possible in the secondary packaging including two oxygen absorbers placed next to the port systems of the two compartments (Figure 1). Fast-acting oxygen absorbers that do not need to be activated by means of sterilization are used so that the absorption of oxygen can start directly after the packaging process. An indicator of integrity of the secondary packaging is also included for proof of sterility and functionality of the absorbers.

## Figures and Tables

**Figure 1 nutrients-12-01849-f001:**
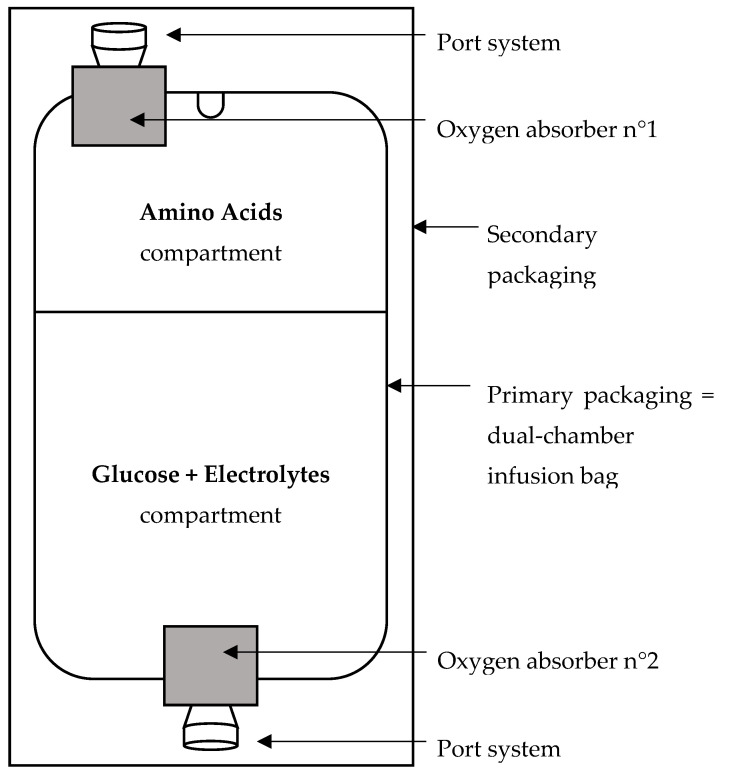
Schematic representation of the packaged dual-chamber infusion bag.

**Figure 2 nutrients-12-01849-f002:**
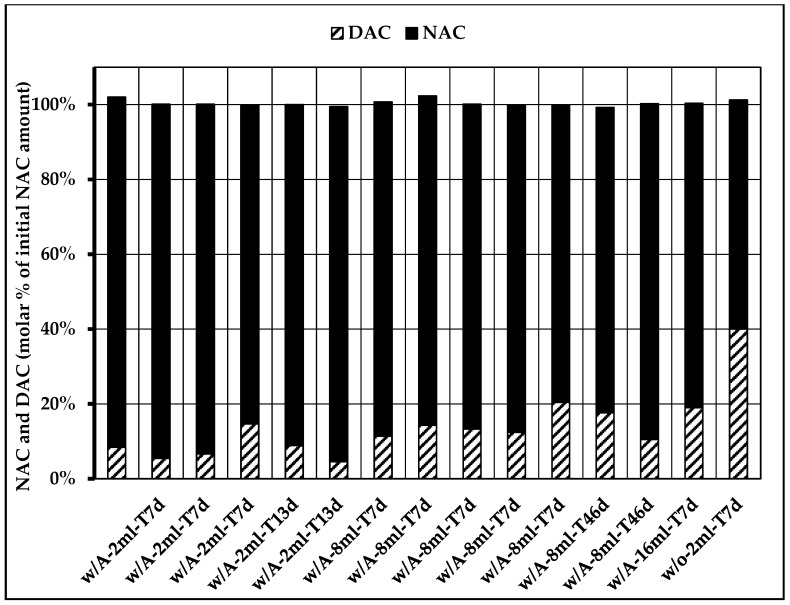
Mass balance for *N*-acetylcysteine (NAC) consumption and *N*,*N*-diacetylcystine (DAC) formation for different storage and manipulation configurations (w/A = with oxygen absorber, w/o = without oxygen absorber, XmL = air volume (21% O_2_), and TYd = days of storage duration).

**Figure 3 nutrients-12-01849-f003:**
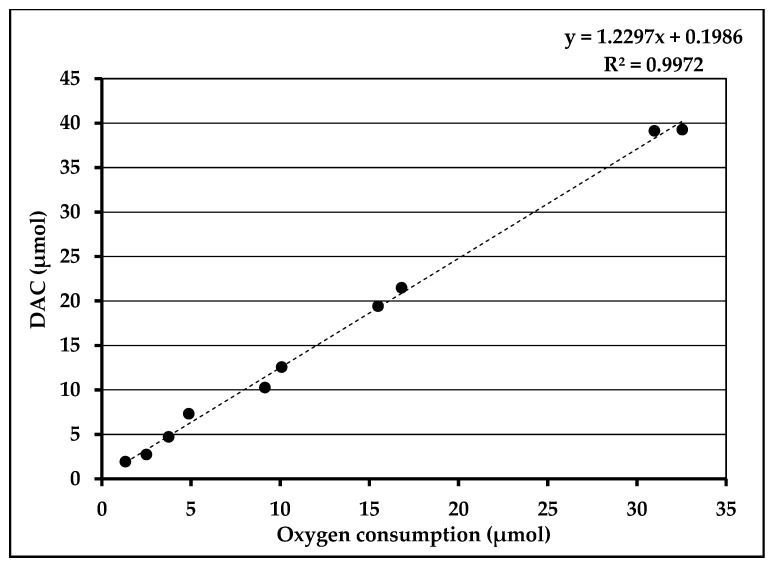
*N*,*N*-diacetylcystine (DAC) formation in correlation with oxygen consumption.

**Figure 4 nutrients-12-01849-f004:**
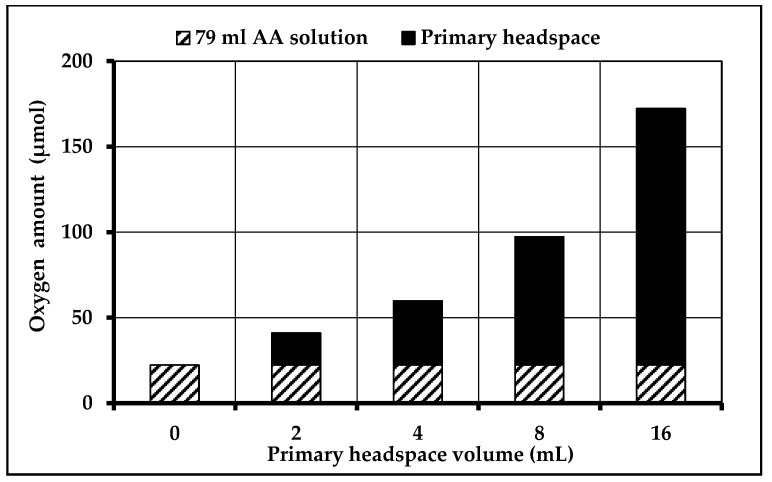
Total oxygen amount within the filled amino acid (AA) compartment depending on the primary headspace (HS1) volume.

**Figure 5 nutrients-12-01849-f005:**
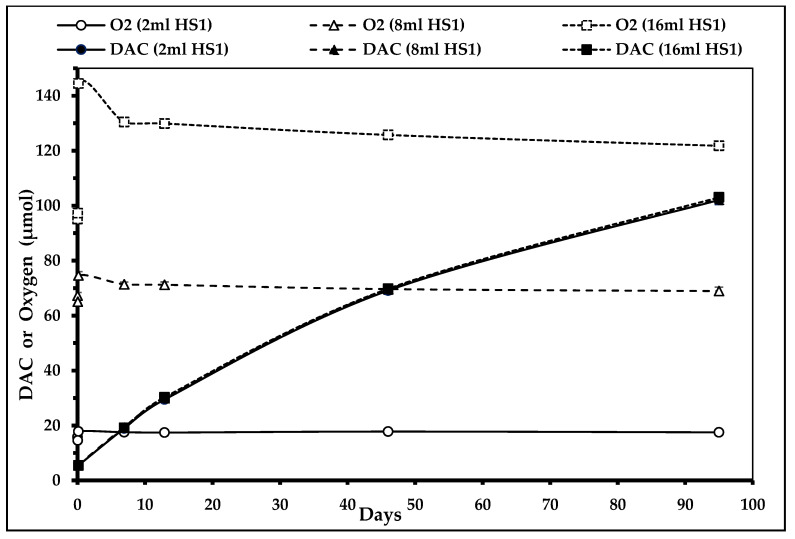
*N*,*N*-diacetylcystine (DAC) concentration depending on primary headspace (HS1) volume in the absence of oxygen absorbers (*n*_2mL_ = 3, *n*_8mL_ = 4, and *n*_16mL_ = 1) (curves for DAC (2 mL HS1), DAC (8 mL HS1), and DAC (16 mL HS1) are superposed).

**Figure 6 nutrients-12-01849-f006:**
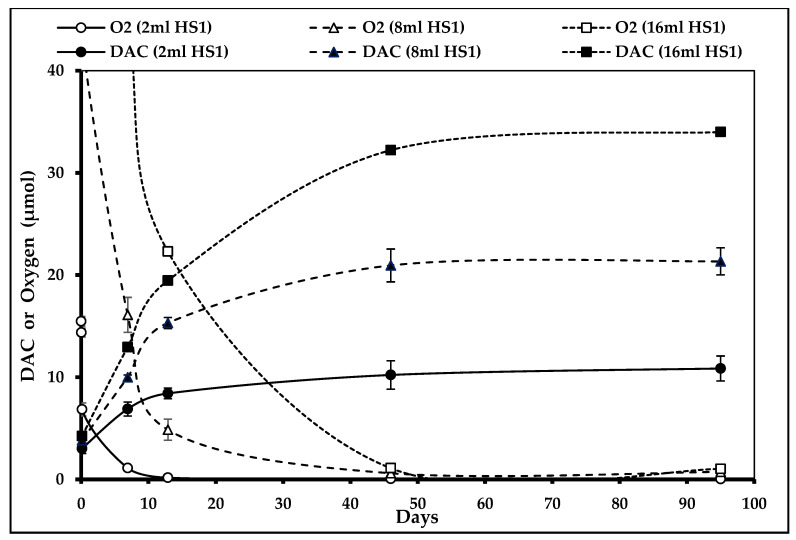
*N*,*N*-diacetylcystine (DAC) concentration depending on primary headspace (HS1) volume in the presence of oxygen absorbers (*n*_2mL_ = 3, *n*_8mL_ = 4, and *n*_16mL_ = 1).

**Figure 7 nutrients-12-01849-f007:**
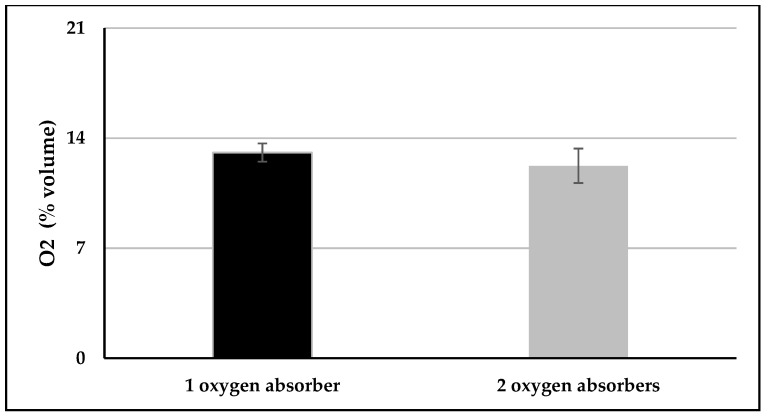
Residual oxygen amount within the primary headspace HS1 after 24 h, starting at 21% of oxygen.

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
