# Peer review of "Stability of N-Acetylcysteine (NAC) in Standardized Pediatric Parenteral Nutrition and Evaluation of N,N-Diacetylcystine (DAC) Formation"

_nutrients, 2020, doi:10.3390/nu12061849_

Round 1

Reviewer 1 Report

The manuscript deals with an essential topic in the field of pediatric parenteral nutrition, especially in neonatal age. The supply of cysteine in pediatric AA solutions is critical due to the immaturity of the enzyme systems in cysteine biosynthesis for preterm neonates and newborns. The instability of cysteine in solution prevents its addition, and therefore it is generally preferred to use N-acetyl cysteine (NAC), which is stable in solution. However, NAC has low bioavailability, and this study tries to demonstrate that the low availability of cysteine is due to the conversion of NAC into N, N-diacetyl cystine (DAC, unusable as a cysteine source) in the presence of high quantities of oxygen.
The methodology adopted in this study seems appropriate, and the conclusions seem supported by results.

However, there are some crucial points that authors need to add, and some corrections to make.

1)In the main text, it is not clearly explained why L-cysteine is not directly added to pediatric AA solutions. It may seem like a solubility problem; instead, it is an instability problem; in fact, cysteine-HCL is very soluble.

2)It is not mentioned that in addition to the AA solutions using NAC (cited in the main text), other solutions use cysteine-HCL (Primene 10% Baxter and Trophamine 10% BBraun).

3)Most importantly, it is not reported that NAC prevents the instability of cysteine in solution but has a 50% bioavailability (ESPGHAN / ESPEN / ESPR guidelines on pediatric parenteral nutrition: Amino acids, page 5).
"The 2006 Cochrane analysis indicated that plasma levels of cysteine were significantly increased by cysteine supplementation but not by N-acetylcysteine supplementation."
In my opinion, this should be reported.

4)The numbering of the figures must be adjusted. There are two figures 1. Figure 2 is shown as Figure 1.

5)Figure 5 is incomplete. The curves for DAC 2 and 8 ml HS1 on the graph are missing. DAC 8 and 16 ml HS1 are missing in the legend.

6)Figure 6 is incomplete. DAC 8 and 16 ml HS1 are missing in the legend.

Author Response

Dear reviewer 1,

Thank you for your constructive comments on my article. I hope having resolved all open points.

  1. More details concerning the different sources of cysteine have been added as well as the reasons why they may not be used or the difficulties to use them. à Line 47-55
  2. Information on AA solutions containing cysteine-HCl have been added. à Line 49-53
  3. NAC bioavailability and the 2006 Cochrane analysis have been reported. à Line 62-64
  4. The numbering of the figures has been adjusted.
  5. Legend of Figure 5 was hidden behind the graph. Figures have been rearranged to show all information. I am sorry for this mistake. The curves for DAC 2 and 8 ml HS1 on the graph are superimposed by DAC 16 mL. Concentration of DAC is the same for all three air volumes (2mL, 8mL, 16mL).
  6. Legend of Figure 6 was hide behind the graph. Figures have been rearranged to show all information. I am sorry for this mistake.

Best regards

Reviewer 2 Report

This is a straightforward study evaluating the conversion of NAC to DAC with exposure to oxygen under varying conditions typically experienced in clinic. The objective is sound and the topic is relevant to the field. The conclusions from the data provide important information, but the presentation of the data are somewhat confusing and there needs to be some statistical evaluation to justify these conclusions.

Main Comments

Line 42-43: Cysteine is always considered a semi-essential or conditionally essential AA given its synthesis depends on an essential AA. What you mean to say is “the semi-essential AA cysteine becomes essential because of low conversion from methionine.”

The methods need to clearly indicate whether these analyses were repeated and clearly state the statistics used to compare data. There are many statistical terms in the paper (increase, higher/lower, no change, etc), but no statistical methods listed nor any indication of statistics performed in the data presented. Without statistics, there is no way to know if the discussion is valid.

Results: Most of the data need error bars to reflect the precision of the analyses. Otherwise there is no way to claim differences between groups without an idea of data variability.

Figure 4: Unclear presentation of data. Do these bars represent O2 content of AA solutions? What does the black vs hatched bars represent—O2 in these compartments? The inference is that there is more O2 in the AA solutions with more headspace, but the hatched bars do not increase with headspace? Need to present these data more clearly with a detailed legend.

Figure 5: What are the solid squares? Are these supposed to solid circles? What exactly is superimposed? There are 4 symbols in the legend, yet 5 symbols in the graph? Why are there multiple symbols at zero time point, yet the lines do not start at their mean? More explanation is needed for this graph.

Figure 6: What are the solid squares? Solid triangles? Many more symbols in graph than in the legend. Was there curve fitting done here? Where are the methods for the curve fitting approaches?

Line 153: No difference based on what stats. Only some of the data seem to have error bars and there is no statistics section, so not sure how this statement is valid?

Line 172-175: Confusing sentence. Please re-word as it’s not clear what point is being made.

Line 198-204: No need to re-state the rationale of the study for the conclusions. Please just state final conclusions from the data.

Minor Comments
Line 110: …data were…

Line 119: …mass balance.

Line 123: Figure 2

Author Response

Dear reviewer 2,

Thank you for your constructive comments on my work. I hope I could answer to all points that had to be more detailed. 

Main Comments

  1. Line 42-43: Some sources say cysteine is a non-essential AA, but as we are talking about PPN, I changed it to a conditionally essential AA as also mentioned in the ESPGHAN guideline on AA. Line 40-46.
  2. More information on statistical validation of the analytical methods has been added. Line 125-135; Repetition of analyses have been explained and more detailed. Data were analyzed by comparing the curves including the error bars. As they were visibly distinct and no overlapping occurred, no further statistical comparisons have been performed. Line 140-144; I hope all statistical terms (increase, higher/lower, no change, etc) are now explained and clear.
  3. Error bars have been added for figures 5 and 6, but most of the time they are not visible. For the 16 mL HS1 curves, no error bars can be added as only one measurement for each time point has been performed.
  4. Figure 4: More information has been given and a detailed explanation of the result has been written. Line 194-197
  5. Figure 5: Legend of Figure 5 was hide behind the graph. Figures have been rearranged to show all information. I am sorry for this mistake. The curves for DAC 2 and 8 ml HS1 on the graph are superimposed by DAC 16 mL. Concentration of DAC is the same for all three air volumes (2mL, 8mL, 16mL). We had three different measurements at time point zero, the first just before injection of air, the second right after injection of air and the third 2.5 hours after injection of air. Therefore, we have multiple points around time point zero. More explanation of the graphs was added. Line 207-214
  6. Figure 6: Legend of Figure 6 was hidden behind the graph. Figures have been rearranged to show all information. I am sorry for this mistake. No curve fitting was done.
  7. Figure 7 has been added to demonstrate the absence of difference between 1 and 2 absorbers. Line 238- 247
  8. Sentence was reworded, I hope it is clear now. Line 251-253
  9. Line 198-204: Final conclusions from the data has been stated. Line 278-284

Minor Comments: have all been modified

Line 110: …data were…

Line 119: …mass balance.

Line 123: Figure 2

Best regards